# Design of a FlexRay/Ethernet Gateway and Security Mechanism for In-Vehicle Networks

**DOI:** 10.3390/s20030641

**Published:** 2020-01-23

**Authors:** Trong-Yen Lee, I-An Lin, Ren-Hong Liao

**Affiliations:** Department of Electronic Engineering, National Taipei University of Technology, Taipei 106, Taiwant105368046@ntut.org.tw (R.-H.L.)

**Keywords:** internet of vehicles, FlexRay, Ethernet, gateway, security, FPGA, power consumption

## Abstract

Due to the development of the Internet of Vehicles (IoV) and advanced driver-assistance systems (ADAS), the Ethernet has become one of the most important communication protocols for the future of automotive networks. This is because the existing communication protocols (such as FlexRay) do not provide sufficiently high bandwidth requirements. The main challenge for the automotive industry will be to transfer and extend standard IP and Ethernet into vehicles and still fulfill the automotive requirements. The automotive gateway not only links two or more protocols and exchanges the data using each, but also monitors and ensures functional safety. This paper proposes a FlexRay/Ethernet gateway by considering the development conditions of embedded systems and the security in the field of vehicle networking. The proposed method is implemented on the Field Programmable Gate Array (FPGA) system to evaluate running time and to analyze the overhead of the security mechanism. For one-to-one mapping logic, the execution times of FlexRay to the Ethernet path and Ethernet to FlexRay path are constant, at 4.67 μs and 6.71 μs, respectively. In particular, cybersecurity can be integrated as an extension of the gateway with low latency and power consumption.

## 1. Introduction

FlexRay [1] is a standard of in-vehicle network (IVN) communication systems that provides high-speed serial communication and is often used for new safety features. FlexRay supports a time-triggered scheme and an optional event-triggered scheme. The advantage of event-triggered communication is better bandwidth utilization, since communication only occurs if needed. The upper bound of the data rate is 10 Mbps, and it provides two channels for redundancy. The Internet of Vehicles (IoV) is a new concept that is an extension of the Internet of Things (IoT). The IoV allows valuable information from other vehicles or the environment to be obtained. An increasing number of electronic control units (ECUs), sensors, and cameras will be equipped in the vehicle for driver-assistance applications. It has been observed that the security and reliability of in-vehicle communication are of paramount importance because malicious attacks can endanger human lives. Therefore, both higher bandwidths and increased security are required.

Ethernet is an important communication protocol, and it has several advantages, such as low cost, high bandwidth, and mature technology. An Ethernet-based vehicle network can communicate between a vehicle and other vehicles. Therefore, Ethernet must become one of the main protocols for IVNs and the IoV in the future [2]. However, the existing Ethernet protocol must be modified to support the delivery of time-critical traffic information and time-sensitive functionality for IVNs. The gateway is an important and indispensable component for IVNs and the IoV. It means that for the introduction of Ethernet, a gateway to currently existing automotive networking technologies, is required for achieving a smooth migration. Therefore, a gateway is an essential component for communication between FlexRay and Ethernet and also helps to further strengthen security. In this paper, in addition to discussing the implementation of the FlexRay/Ethernet gateway system, we also analyze the cost of the system security constraints on latency and power consumption.

## 2. Related Works

### 2.1. Gateways

In the automotive industry, embedded system architectures are set in vehicles. The gateway in IVNs is designed to check internal and external messages and route data from one control protocol to another. There are two kinds of vehicular network structures: centralized gateway and backbone network. In a typical centralized gateway structure, communication protocols IVNs are interconnected by a central gateway. The central gateway is usually implemented by processor to perform the tasks of packing and unpacking messages, and lookup [3]. The scalability of the centralized gateway is related to the switching mechanism. In the routing (switching) table, the valid IDs (devices) with priority and destination addresses are listed. When the number of ports is increased, the routing table will be expanded, and there will be some overhead costs in terms of latency and hardware resources.

First, an accelerated routing mechanism using a Field-Programmable Gate Array (FPGA) to reduce the latency for a gateway was proposed in [4]. The authors also built several demonstrations with different bus systems. They proposed that a modular gateway prototype based on an FPGA is a powerful alternative for automotive gateways. To minimize the bandwidth requirements, a scheduling algorithm, earliest deadline first (EDF), was proposed in [5]. EDF forwarding was able to achieve real-time capability using fewer resources. In addition, the authors adopted a frame aggregation mechanism to reduce the overhead of framing.

Recently, many studies have been published on the vehicular gateway [6,7,8]; comparisons of platforms and functions of security are shown in Table 1. A variety of gateways [6] have been proposed that use heterogeneous network protocols, including CAN, FlexRay, and Ethernet. The authors proposed a reusable, verifiable gateway framework that could be easily developed and verified using graphical user interface (GUI)-based configuration software. However, the verification environment provided by the proposed gateway framework has not yet been verified. A configurable low-latency gateway architecture on a hybrid FPGAs was proposed in [7]. A translation mechanism based on a virtual MAC address was used. The virtual MAC address corresponds to a data type and a set of receivers that are pre-loaded into the lookup and packing/unpacking module together with the FlexRay message-type identifier. Furthermore, every slot also contains a preemption indicator (PI), which allows preemption of long messages by high-priority messages and can be used to implement Ethernet priority schemes, such as time-sensitive networking. A synchronization mechanism was developed in [8] for FlexRay and Ethernet AVB networks that guarantees a high quality of service (QoS). The authors also designed a corresponding embedded system-based gateway system to implement the proposed synchronization mechanism. This gateway, which is an Ethernet-backbone network, will be used in future automotive systems.

However, these methods have weaknesses in terms of cybersecurity. The two major objectives of gateways for IVNs and the IoV include a conversion device between different protocols and a supervisory role to ensure the packets are correct, and a gateway host to control and monitor the gateway system that is aimed at improving the security of the Ethernet/FlexRay gateway. We previously proposed a gateway system between the Ethernet and FlexRay in [9]. In this current paper, we describe the proposed gateway system in more detail, providing information about each module of the gateway host, risk-control strategies, and the embedded system used for implementation of the proposed gateway.

### 2.2. Automotive Ethernet and Security

Next-generation automotive systems will provide more advanced applications; therefore, extending the Ethernet into vehicles will bring many benefits. Integrating Ethernet for in-vehicle communication, reliability, and security are the main concerns [10]. Because Ethernet was not originally designed to be a real-time system, deploying Ethernet into vehicles requires additional features to the original protocols to ensure safety and reliability [11]. Along with the increasingly complex interfaces and automotive Ethernet to connect IVNs to the outside, dangerous attacks can be easily implemented on IVNs. IVN security is essential, and the topic is of interest in the field.

The security concept for vehicular gateways is designed to provide security to vehicles and to ensure individual privacy in a heterogeneous network. For example, vehicles could exchange useful messages such as road conditions and accident notifications in order to assist in safe navigation and traffic management. The nodes in such critical applications are very vulnerable to vicious attacks; the Sybil attack can be classified as one of the most dangerous attacks on the IoV [12]. In this scenario, a vehicle can appear to have more than one identity. In other words, other vehicles in the network are unable to determine the information originator. One type of Sybil attack is the Node Impersonation Attack [13]. Each vehicle’s IVN has a unique identity, and vehicles use their identities while communicating with other vehicles. However, if a vehicle changes its identity without informing the managing unit, it can introduce itself as a different vehicle, such as in a Sybil attack. The malicious vehicle can send incorrect information about the road conditions to the surrounding managing unit. Recently, research has focused on security risks to IVN protocols, such as CAN [14]; few studies have investigated detection and prevention mechanisms on FlexRay. The targets of an automotive gateway are seamlessness and improved security. Therefore, we designed the gateway host on the FPGA hardware to process data across these heterogeneous networks with low latency and increased cybersecurity awareness.

## 3. Proposed Methodology

The functionality of the gateway is to play a role similar to an interface between a FlexRay bus and an IEEE 802.3-based Ethernet network. The design of an automotive FlexRay/Ethernet gateway system is shown in Figure 1. The concept of this system is to implement the data-packing and data-unpacking FlexRay messages in Ethernet packets, a transmission control protocol/internet protocol (TCP/IP) stack, and an Ethernet controller. 

In general working conditions, the Ethernet utilizes specific addresses (the MAC addresses) to identify the destination and source, and a transformation based on a virtual MAC address that corresponds to a data type and a set of receivers with the FlexRay message type identifier is used. However, in more concrete cases, attack and overloading problems always happen in the heterogeneous network. When a malicious vehicle (non-valid device) wants to provide incorrect messages, the switching mechanism will reject the messages. Additionally, when messages collide on the network, the processing is done according to the priority queue in the routing table.

### 3.1. System Architecture

The architecture of the proposed FlexRay/Ethernet gateway and data path is shown in Figure 2. It is composed of three units: the FlexRay controller, the Ethernet controller, and the Gateway host. FlexRay and Ethernet communication are essential for gateway implementation. The gateway host is designed to load the four modules to receive data and translate the Ethernet event frame to the in-vehicle system event and vice versa. The secure data transmission mechanism (SDTM) is used to further improve the security of the gateway system.

### 3.2. The Gateway Host

The gateway host consists of four modules, which are a Receive module, Security module, Transform module, and Transmission module, with a secure data transmission mechanism (SDTM) to further improve the security of the system. Moreover, the SDTM is divided into two parts, one for the Security module of the Gateway Host and another for the encryption of the Ethernet frame. The original protocol of the FlexRay frame and Ethernet frame have been some check mechanisms, such as CRC and UDP/TCP header check, which can be adopted in the proposed SDTM and is a part of the gateway. The CRC checks are implemented in the FlexRay controller and Ethernet controller, respectively. The UDP/TCP header check mechanisms are implemented in the TCP/IP networking stack. When the security verification results fail, the gateway host will drop an incoming frame. The main function module of the gateway host is described as follows: In the receive module, there are two different buffer registers of FlexRay and Ethernet in the module. The registers buffer frames from the Rx bus and extract the important parts of the Header segments, the Payload data segments, and the Trailer segments of the frames.

The security module is a part of the SDTM; there are four check mechanisms in the proposed Security module. First, there is a cyclic redundancy check (CRC) to ensure that safety measures on the data link layer integrate FlexRay and Ethernet bus technologies. Both consist of protection of header and payload information as part of each frame using checksums, such as the CRC. Second, if Ethernet is used in combination with the IP, additional mechanisms are incorporated to aid in security concerns. Part of the IP header information (e.g., Source Address, Destination Address) is protected by a 16-bit checksum. In Ethernet architectures, real-time data is exchanged by the widely used User Datagram Protocol (UDP) in addition to the IP. UDP adds an additional 16-bit arithmetic checksum to protect the underlying IP header. Both the sequence number and the acknowledgment number of the TCP header are necessary roles in the three-way handshake of TCP protocol operations. Therefore, these error detection mechanisms are adopted in the proposed Security module. 

Third, the sequence number of the TCP header has dual roles. In the initial sequence number, the synchronization (SYN) flag is set to 1 in the TCP header. The actual first data byte of the sequence number and the acknowledged number in the corresponding acknowledgement (ACK) are then this sequence number plus 1. If the SYN flag is clear (i.e., 0), then this is the accumulated sequence number of the first data byte of this segment for the current session [15]. Fourth, the acknowledgment number of the TCP header can also be adopted in the proposed check mechanism. If the ACK flag is set, the value of this field is the next sequence number that the sender is expecting. If this is the case, this acknowledges receipt of all prior bytes. The first ACK sent by each end acknowledges the other end’s initial sequence number itself, but no data.

The central component of the gateway architecture is the transform module, transforming the frame tasks to each other. The methods of the FlexRay/Ethernet frame [1,16] transformation are shown in Figure 3 and Figure 4. The transform module consists of the following three segments: header-generated logic, payload data queue logic, and trailer generate logic. The data path from FlexRay to Ethernet is shown in Algorithm 1. Since the length of Ethernet frames is more than ten times that of the FlexRay frames, the Transform module adopts one-to-one mapping logic to encapsulate the frames. The verification of the security mechanism is status-receiving process, frame CRC, and (FlexRay) ID authentication. On the other hand, the purpose of encryption of the Ethernet frame is to protect the privacy of secure data during communication with external devices or further prevent attacks. The encrypted message from the network must be read and decrypted using the current configuration of the cipher primitives before the information can be used by the application in the Ethernet node. In the other case, the messages are from Ethernet to FlexRay, as shown in Algorithm 2. The Transform module must divide the payload segment from an Ethernet frame into several frame packages (maximum of six) with the same header and trailer. The message identifier is directly based on the message received, and the length of Ethernet data is double that of FlexRay data. The virtual MAC address corresponds to the data type and the FlexRay ID parameter forming one-to-one mapping to a virtual MAC address.
**Algorithm 1.** Data path from FlexRay to EthernetInputs: Frame of FlexRayOutputs: Frame of Ethernet1:Initialize all registers2:Establish TCP connection between Gateway and Ethernet End3:**While** (FlexRay frame received) **do**4:Fetch ID, Length, and Data in FlexRay frames and storage to buffers5:Send ID, Length, and data to Security module6:*if* (ID is found in the routing table *and* security verification checks pass) {7:According to the Routing table, the SA and DA is set8:Set length of Ethernet = (length of FlexRay) × 29:Set Ethernet_Data = FlexRay_Data*/**encryption processing of SDTM**/*10:Encryption of the data field in the Ethernet frame11:Set Ethernet controller frame buffer12:Send Ethernet frame to Ethernet End13:}14:else{/* security verification failure*/15:The Gateway host drops the frame16:**Return**17:}18:**End While**

**Algorithm 2.** Data path from Ethernet to FlexRayInputs: Frame of EthernetOutputs: Frame of FlexRay
1:Initialize all registers2:Establish TCP connection between Gateway and Ethernet End3:**While** (Ethernet frame received) **do**4:Fetch SA, DA, Length, and Data in Ethernet frames and storage to buffers5:Send SA, DA, Length, and Data to security module6:*if* (SA is found in the routing table and security verification checks pass) {7:Set the FlexRay_ID according to the routing table*/** divides the payload segment from Ethernet frame into several frames* */*8:*if *(length of Ethernet > 255 Bytes) {9:Divide the Data into several Data and calculate length of each Datum10:Calculate the number of divided data (***n***)11:}12:else{/* security verification failure*/13:The Gateway host drops the frame14:
**Return**
15:}16:Generate Header CRC17:Forach (***n***)18:FlexRay_Data = Divided (Ethernet_Data)19:Set FlexRay controller frame buffers20:Send FlexRay frames to FlexRay Node21:}22:
**End While**



The transmission module packs the parts from the transform module into new frames, buffers the frames, and schedules the output frames to the Tx bus according to their priorities. The protocol data unit (PDU) is useful for network protocols that support a longer frame length than CAN, such as FlexRay and Ethernet. In the case of Ethernet, a frame can include multiple PDUs because the minimum frame length of the Ethernet is too long to store single data. PDU-based routing consists of direct and indirect routing. Direct PDU-based routing immediately transmits a received PDU to the destination network, but it can route part of the frame to a destination network without copying the entire frame. A received PDU that is configured to direct PDU routing is immediately forwarded to the destination interface module, such as the CAN or FlexRay interface.

### 3.3. Implementation of the Proposed Gateway

The automotive gateway links two or more protocols and exchanges the data between protocols. Therefore, there are at least two protocol controllers in the gateway system. The gateway integrates those controllers, and the data transmits components or buses. The proposed gateway system is implemented on a Xilinx Zedboard, as shown in Figure 5. The Zedboard is a heterogeneous system architecture; it integrates Programmable Logic (PL) and Processing System (PS) with ARM9 cores. The proposed gateway host and FlexRay controller are implemented on PL fabric, and the Ethernet node is implemented by both PS and PL.

The FlexRay node is composed of the FlexRay controller and FlexRay transceiver. The FlexRay controller adopted in the proposed gateway system is intellectual property developed by the PATAK Engineering Company [17]. We modified the source code (hardware description language, i.e., Programmable Logic) for experiment planning. There are two communication paths (Receive path and Transmission) between the gateway and the FlexRay controller. The interconnection of the gateway control *I*/*O* port and the FlexRay controller are internal ports, such as Data input, Data output, and clock (CLK). The external ports such as Tx, Rx, and Tx_Enable are connected via the FPGA Mezzanine Card (FMC) pins on the ZedBoard for external communication. Furthermore, we designed and implemented a FlexRay transceiver board to connect the external transceiver to verify the proposed gateway system with the ZedBoard. Therefore, the FlexRay controller directly connects to the gateway and transmits the FlexRay data to the transceiver with FMC ports.

In the Ethernet node, the PL fabric used implement a 10/100/1000 Ethernet port for a network connection using a Marvell 88E1518 PHY (Physical), and the PHY interfaces to the Zynq-7000 AP SoC via an RGMII. A simple TCP/IP stack is implemented on the Ethernet controller to handle the Ethernet data with the Marvell PHY, and the AXI buses are used to communicate with the gateway, which is also on PL fabric. In this system, the TCP/IP can be defined as a four-layer model, as shown in Figure 6. The Application layer, Transport layer, and Internet layer are implemented using the LwIP Application on a PS. The Lightweight IP (LwIP) [18] is an open-source TCP/IP networking stack for resource-limited systems, such as embedded systems. The Link layer is composed of a MAC layer and a Physical layer, which are implemented by a Gigabit Ethernet MAC Controller on a PS and an Ethernet PHY interface on PL, respectively.

### 3.4. Design of the Secure Data Transmission Mechanism

There are several intrinsic vulnerabilities with IVNs [19]. Owing to these vulnerabilities, adversaries can easily implement various attacks on IVNs, including frame sniffing, frame falsifying, replay attack, frame injection, and DoS attack. Corresponding countermeasures can be used to protect the IVN from these kinds of attacks, such as encryption and authentication, development of the Intrusion Detection System (IDS) [20], and separation of potential attack interfaces from the IVN. Building the lookup table for the virtual MAC and the Ethernet MAC is a lightweight authentication scheme for IoV. The security mechanism will drop the packet from a malicious vehicle with a non-valid ID; therefore, the incorrect messages are excluded to avoid these attacks.

To ensure information security, the protection mechanisms have to be integrated into the IVN. The proposed secure data transmission mechanism (SDTM) is intended to enhance IVN security by encryption and authentication. The proposed SDTM is divided into two parts: one for the encryption processing of the Ethernet frame and another for the Security module of the Gateway Host. On the other hand, this paper identifies three applicability criteria, which show the security levels and applications in vehicle systems of the risk-control strategies. In what follows, we explain these criteria and discuss their consequences; an overview of the proposed applicability criteria and the corresponding Security mechanism for risk-control strategies is shown in Table 2. 

Risk-control strategies

There are multi-level security strategies in the proposed method based on the application of FlexRay data signals. Multilevel security protocols were also adopted in [21]. Wang et al. noted that adding security mechanisms can protect the IVN by collaborating with current in-vehicle security schemes. These discrepancies were made according to the applications in in-vehicle systems. At the highest level, all of the security mechanisms are enabled, and the applications are often used for new safety features, such as X-by-wire [22] and powertrain systems. When the proposed gateway is used in the collision avoidance system or electronic stability control, the security level is set to medium, and the checking of the MAC address and ID mechanism is disabled. In the low-security level mode, only the checking states of receive interfaces and checking frames of CRC mechanisms are enabled. The flow chart of the security module is shown in Figure 7. The processing of the checking UDP header is implemented in the LwIP stack only applicable in the Ethernet to FlexRay path, and the packets are dropped upon verification failure.

In such cases, when the gateway receives a frame from FlexRay or Ethernet, the status-receiving process is an error; when the gateway receive a frame from FlexRay or Ethernet, the frame CRC is not correct; when the gateway receives a frame from Ethernet, the sequence number and acknowledgment number of the TCP header, which is related to TCP protocol operation, is an error; when the gateway receives a frame from FlexRay, the MAC address is not in the whitelist. Then, the receiving process will drop an incoming message and no further processing is required.

Encryption methods

Although currently most IVNs do not use encryption, this might be a very important aspect in the future [23]. There are three encryption methods that are used in the proposed SDTM: Data Encryption Standard (DES), Advanced Encryption Standard (AES), and AES Counter with CBC-MAC (AES-CCM). A comparison of encryption methods is shown in Table 3.

DES is a classical symmetric-key algorithm for the encryption of electronic data with a 56-bit key size. The operation speed of encryption is the fastest of the three encryption methods, but the DES key is easy to break in a short period of time.AES supersedes the DES with a 128-bit key size. It is a subset of the Rijndael block cipher. Research into attacks on AES encryption has continued since the standard was finalized in 2000, which means the security is sufficient for modern vehicle communications.AES-CCM is one of the authenticated encryption schemes as specified by the National Institute of Standards and Technology (NIST) [24]. The CCM mode is a mode of AES operation for cryptographic block ciphers and is used to protect the static segment of FlexRay, which is recommended by the IEEE 802.11 standard. Thus, the AES-CCM is considered the future encryption method in the IoV [25].

## 4. Experimental Results

The performance of the proposed FlexRay/Ethernet gateway is evaluated in this section. The process of transforming FlexRay and Ethernet frames was proven by the experiment results.

### 4.1. Experimental Platform and Environment

The experimental environment and EDK tools with the proposed gateway architecture are shown in Table 4. The proposed gateway was implemented using a Xilinx ZedBoard (Zynq XC7Z020-1CLG484). The ZedBoard implemented a 10/100/1000 Ethernet port for network connection using a Marvell 88E1518 PHY and two Gigabit Ethernet Controllers (10/100/1000 Ethernet MAC compatible) on PS fabric. The whole ZedBoard connection used to implement a gateway is shown in Figure 8.

### 4.2. Execution Time Management and Analysis

To measure the execution time of the proposed gateway system, we considered the verification tools and transmitting messages. The Network Debug Assistant was applied to build the TCP communication (client or server on PC) with the LwIP via an RJ45 cable. The Network Debug Assistant not only has a three-way handshake process, which is necessary for TCP communication, but we could also utilize it to send packets to the ZedBoard to evaluate the performance of messages from the Ethernet to the FlexRay path. On the other hand, the Wireshark [26] network protocol analyzer tool was used to capture, filter, and inspect the Ethernet packets that were sent from the ZedBoard to evaluate the messages from the FlexRay to the Ethernet path.

Furthermore, we employed an 8-byte message in the evaluation of the FlexRay ECU to an Ethernet network. This feature, according to the related work, showed that the 8-byte message size represented more than 70 percent of the traffic on FlexRay-based vehicular systems. Multiples of such messages are packed together to form valid Ethernet payloads of 64 bytes. The controller of the hardware-based gateway has a fetch-and-pack task that is activated whenever an 8-byte FlexRay frame is received at the network interface. The task reads the message into the Ethernet buffer if the packet is ready to be transmitted; otherwise, it executes other tasks and waits for the next interruption.

The execution times of FlexRay to Ethernet and Ethernet to FlexRay are shown in Table 5. The latency measurement is denoted as follows: the message is transmitted from the FlexRay path to the Ethernet path; the starting is the FlexRay node, which sends out messages; the gateway host receives the FlexRay frame, and then each module starts to extract the data from the frame; the ending point is the Ethernet controller. In the other path, the latency components are Ethernet end (means that the execution time of LwIP application in SDK.), Gateway host, and FlexRay Node.

For these experiments, the proposed gateway host was executed multiple times for every FlexRay/Ethernet frame, consuming considerable processor cycles during the movement of data, and there are 256 valid-IDs in the routing table to one-to-one mapping. The running time of the Ethernet to the FlexRay path in the Gateway host was more than that of the FlexRay to Ethernet path because the payload segment from the Ethernet frame was divided into several frames, as seen in Algorithm 2. Furthermore, it was found that the execution time of the FlexRay to Ethernet path in the Ethernet End was more than that of the Ethernet to FlexRay path; this was because of the encryption (with the AES-CCM method) of the data field in the Ethernet frame, as shown in Algorithm 1. Furthermore, the security model maybe suit for critical situations according to the human response time (200 ms in general).

### 4.3. Cost of Security Mechanism

The proposed SDTM consists of the Security module and the encryption processing of an Ethernet frame. The key challenge is to integrate this complex security architecture in a manner that introduces minimal latency (for the network or application) and without affecting protocol guarantees. In the experiments of the proposed SDTM, we measured the time cost of the encryption processing and then analyzed the overhead of the security mechanism.

As presented in Section 3.4, the methods of DES, AES, and AES-CCM can be adopted in the encryption processing of messages from the FlexRay to the Ethernet path. For security managed through hardware, the encrypted message received from the network must be read and decrypted using the current configuration of the cipher primitives before the information can be used by the application. For a message from the FlexRay to Ethernet path per 8 bytes of data, the encryption methods (DES, AES, and AES-CCM) take 1.17 μs, 1.23 μs, and 3.28 μs, respectively. When we applied the AES-CCM method in the encryption processing, the method used 17.7% of the total execution time of the FlexRay to Ethernet path.

To analyze the overhead of security mechanisms, two cases were implemented: the proposed gateway with and without the security mechanism. While embedding security mechanisms into the proposed gateway systems, there were some overhead costs in terms of latency, hardware resources, and power consumption, as shown in Table 6. Increasing the security level increased the latency due to the complexity associated with management of the check mechanism operations. The results showed that the proposed SDTM can be integrated as an extension of the gateway with low-latency overhead.

### 4.4. Performance Evaluation

Execution time of the proposed FlexRay to Ethernet gateway and the Ethernet to FlexRay gateway were compared with other studies. Execution time includes the execution time of interruption, extraction data frame, data frame transformation, security check mechanism, and transmission process. The results for the running time of the modules are shown in Table 7. A comparison of running time with related studies is shown in Table 8. Both related works [6,7] use multiple nodes for experiments; the proposed centralized gateway adopted the length of data represents 8-bytes of payload data for a one-to-one mapping logic to encapsulate the frame. Kim et al. [6] proposed a layered architecture that was hardware- and software-independent, and that provided parallel reprogramming, diagnostic routing, and network management. A configurable vehicular gateway was proposed by Shreejith et al. [7], which implemented a reconfigurable dedicated switching circuit but did not integrate security primitives.

The proposed gateway included a gateway host to control and monitor the gateway system in order to improve the security of the Ethernet/FlexRay gateway. Although the comparisons of execution time were not ideal, the latency overhead is constant under any conditions, architectural design strategies for cybersecurity and functional safety were implemented. Then, the contributions of this paper are as follows:(1)Using the HW/SW (hardware/software) method, data communication is implemented between the FlexRay and Ethernet network.(2)The risk-control strategies are related to FlexRay data signals, and the proposed Security Data Transmission Mechanism (SDTM) is integrated as an extension of the gateway.(3)The novelty of the vehicular gateway is to improve the security with little overhead, and it is more suitable for IVNs and the IoV in the future.

## 5. Conclusions

In this paper, a gateway for the FlexRay and Ethernet interface design using an FPGA hardware core and a CPU software core is proposed. The proposed gateway is implemented on a Xilinx ZedBoard using IP integration and the HW/SW method. The secure data transmission mechanism (SDTM) including security module and encryption processing is designed for the detection and monitoring of data packets. Three risk-control strategies based on the type of FlexRay data signals are designed for different car applications or regions of the FlexRay protocol. An experimental method to verify performance is conducted to make connections between the proposed gateway and an external FlexRay node or Ethernet end. In the future, the gateway should be implemented on the platform designed to meet the requirements for modern vehicles. Additionally, the evaluation environment should be implemented in automotive software architecture such as AUTOSAR, which is closer to the real automotive environment.

## Figures and Tables

**Figure 1 sensors-20-00641-f001:**
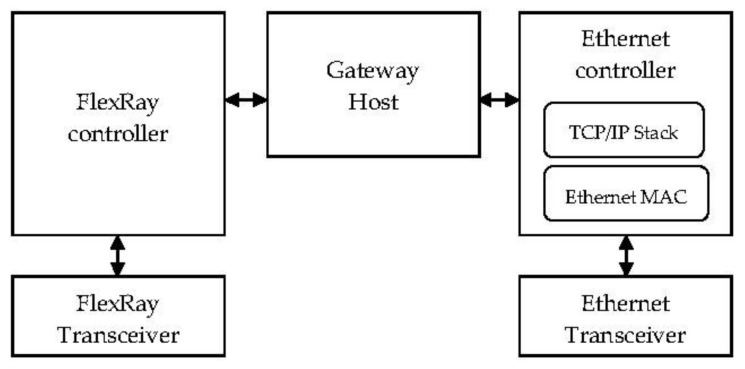
An overview of the implemented system. TCP/IP is transmission control protocol/internet protocol.

**Figure 2 sensors-20-00641-f002:**
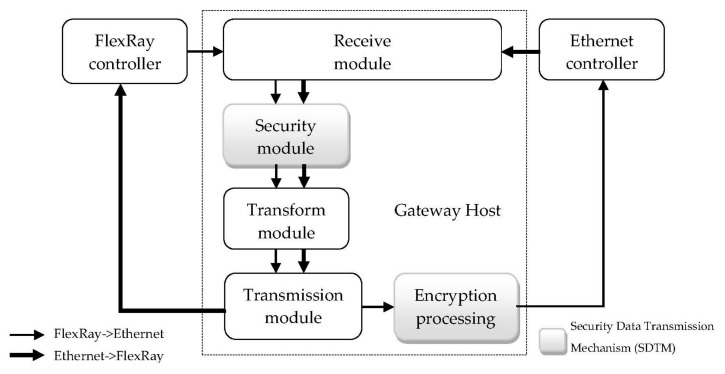
The proposed FlexRay/Ethernet gateway architecture and data path.

**Figure 3 sensors-20-00641-f003:**
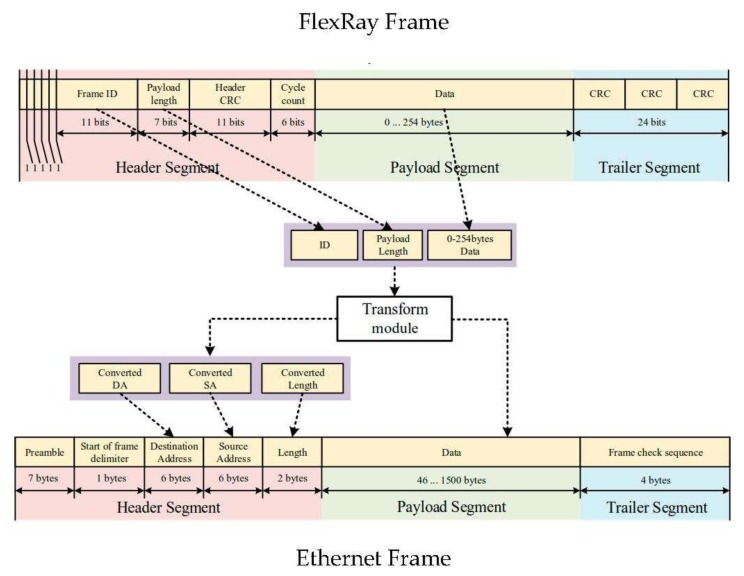
FlexRay frame transformed to Ethernet frame.

**Figure 4 sensors-20-00641-f004:**
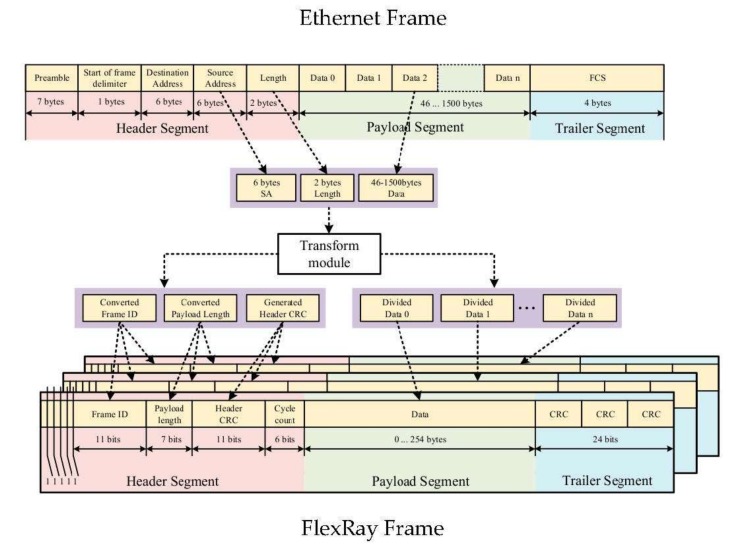
Ethernet frame transformed to FlexRay frame.

**Figure 5 sensors-20-00641-f005:**
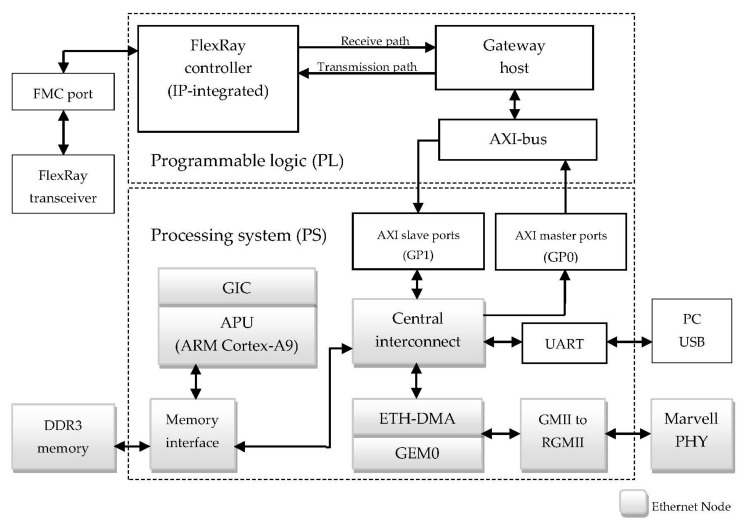
The design of proposed gateway system based on Zynq-7000 AP SoC. The Ethernet node is implemented by both PS and PL.

**Figure 6 sensors-20-00641-f006:**
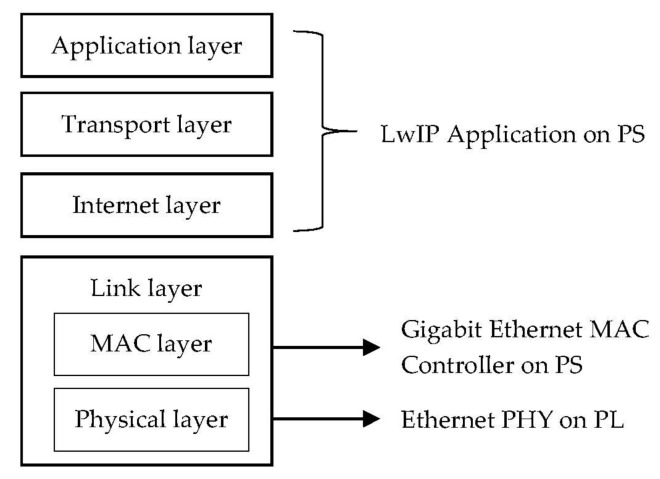
The design of the TCP/IP model on a ZedBoard.

**Figure 7 sensors-20-00641-f007:**
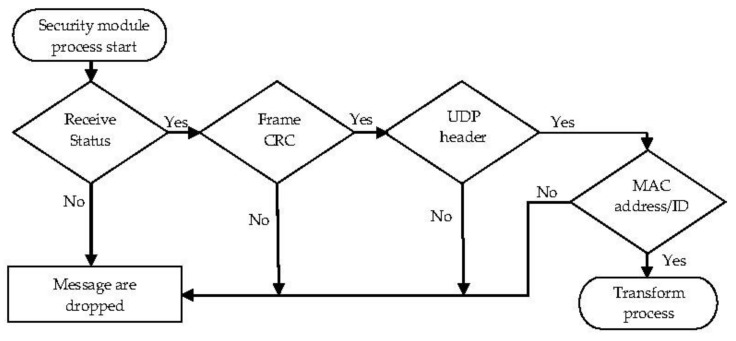
Flow chart of the security module.

**Figure 8 sensors-20-00641-f008:**
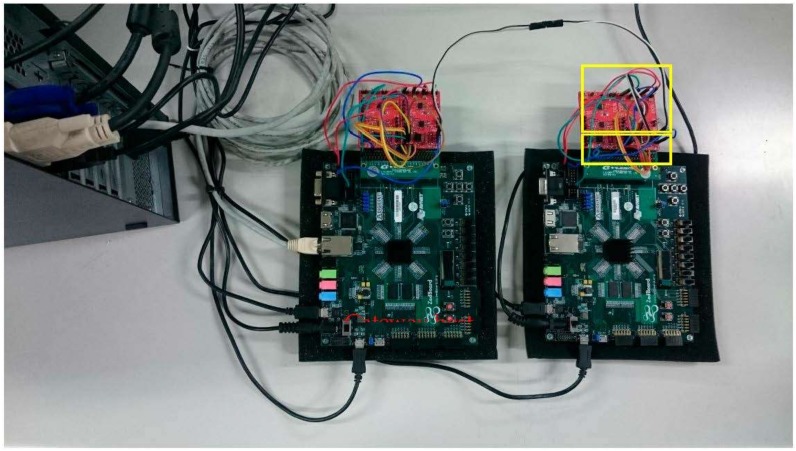
Overview of the hardware platform setup.

**Table 1 sensors-20-00641-t001:** Comparisons of platforms and their security functions.

Method	Development Board	Manufacturer	Platform Clock	Security Mechanism
Kim [6]	MPC5668EVB	NXP	116 MHz	not included
Shreejith [7]	ZC702, ZC706	Xilinx	200 MHz	not included
Lee [8]	TC275EVB	Infineon	200 MHz	not included
This paper	XC7Z020	Xilinx	100 MHz	included

**Table 2 sensors-20-00641-t002:** Risk-control strategies.

Security Level	Applications (Examples)	Security Mechanism			
		Receive Status	Frame CRC	UDP Header	MAC Address/ID
High	Powertrain systems X-by-wire	enable	enable	enable	enable
Medium	Collision avoidance system Electronic stability control	enable	enable	enable	disable
Low	suspension Traction control system	enable	enable	disable	disable

**Table 3 sensors-20-00641-t003:** Comparison of encryption methods.

Encryption Methods	Parameters			
	Data Block (Bits)	Key Length (Bits)	Rounds (Times)	Mode of Operation
DES	64	56	16	N/A
AES	128	128/192/256	10/12/14	ECB-mode
AES-CCM	128	128/192/256	10/12/14	CCM-mode

**Table 4 sensors-20-00641-t004:** Experimental environment.

Proposed Architectures	Implementation Platform/OS	EDK/Tools	Resource Measurement
FlexRay Node	Xilinx Zedboard Zynq-7000 AP SoC	Xilinx Vivado 2018.1	Xilinx Vivado 2018.1 utilization report
Proposed Gateway host	Xilinx Zedboard Zynq-7000 AP SoC	Xilinx Vivado 2018.1 Xilinx ISE 14.7	Xilinx Vivado 2018.1 utilization report
Ethernet End	Windows 7 Professional Intel Core i7 3.6 GHz	Network Debug Assistant Wireshark	N/A

**Table 5 sensors-20-00641-t005:** Execution Time (μs) of the message path.

Messages Path	Latency Components			
	FlexRay Node (PL)	Gateway Host (PL)	Ethernet End (LwIP in PS)	Total Run Time
FlexRay -> Ethernet (with AES-CCM)	8.25	4.67	5.55	18.47
Ethernet -> FlexRay	7.96	6.71	1.36	16.03

**Table 6 sensors-20-00641-t006:** Area and power overheads on the Xilinx Zynq-7000 device.

Implementation Method	Resource Consumption (Reg, LUTs)	Resource (LUTs)	Power Consumption
**Proposed gateway**	1.00× (4950), 1.00× (4880)	9.3%	1.00×
**Proposed gateway** **with SDTM**	1.25× (6125), 1.24× (6182)	11.62%	1.01×

**Table 7 sensors-20-00641-t007:** Detail for execution time (µs) of modules within the proposed gateway host.

Message Path	Receive Module	Security Module	Transform Module	Transmission Module	Total Runtime
FlexRay -> Ethernet	0.86	0.47	2.13	1.21	4.67
Ethernet ->FlexRay	1.14	0.61	3.72	1.24	6.71

**Table 8 sensors-20-00641-t008:** Comparisons of FlexRay/Ethernet gateway execution time (µs).

Message Path		FlexRay -> Ethernet	Ethernet -> FlexRay
Method	Port	Execution Time	Execution Time
Kim [6]	Multiple nodes	67	67
Shreejith [7]	Multiple nodes	3.15	3.33
Proposed	One-to-one mapping	4.67	6.71

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
