# Peer review of "Design of a FlexRay/Ethernet Gateway and Security Mechanism for In-Vehicle Networks"

_sensors, 2020, doi:10.3390/s20030641_

Round 1

Reviewer 1 Report

The article focuses on design of vehicle gateway structure. The topic is vital and interesting for community focused on in-vehicle networking. Nevertheless, I identified the following issues in the article:

The article structure is not good. I recommend to follow the structure of cited work [4]. The gateway structure is not described well - it should be clarified what is implemented in FPGA logic and what in software running on processor. Which way the switching (routing) is implemented? It seems the proposed gateway provides only single Ethernet and single FlexRay port. What about the scalability of the gateway structure, is higher ports number possible? What is the influence of routing table size on the gateway latency. For which routing table size the presented results were measured? The concept of gateway security features is not clear. TCP protocol is mentioned here (e.g. Fig. 6) - it is seldom used for in-vehicle communication, UDP is preferred. What is the role of LwIP stack here? It is SW implementation and if included in data processing, I do not believe in (cca)5us latencies. The latency measurement should be defined precisely - what are the starting and ending points? Is the same approach used in [4]? I do not understand value 8.3% in column "Average" in Table 8. The comparison in Table 9 is "not correct", solution presented in [4] provides 4 gateway ports and switching fabric requires lot of logic. Abbreviation "IDS" is used in chapter 2.4, but not explained before. Deep English language review is necessary, I did not understand some sentencies at all.

Author Response

Responses to Reviewer’s (#1) comments:

Thank you for your time and suggestions. We have modified the manuscript based on your comments as follows.

According to your comment that we have revised the structure of the We modified Figure 5, to clarify what is implemented in PS and what in PL. We added the description about switching mechanism in Section 2.1. We added the description about the scalability of the gateway structure in Section 2.1. We added the description of the influence of routing table size on the gateway latency in Section 2.1. The description of the measurement of routing table size in Section 4.2. We added the description about gateway security features in Section 2.2. We revised the TCP to UDP in Figure 7. We added the description and Figure 6 in Section 3.3 about LwIP stack. The Ethernet controller is implemented by both PS and PL, the execution time of the Ethernet controller means that the execution time of LwIP application in SDK. We added the description about latency components in Section 4.2, and revised the Table 5. We deleted the incorrect comparison in Table 8, and deleted the description of “average”. We deleted the incorrect comparison in Table 9, and deleted the incorrect sentences. The explanation and reference of IDS has been added in subsection 3.4. The manuscript has been revised by extensive editing. In many places, sentences have been completely rewritten. Some miss-spellings in content has been corrected. The manuscript has been checked extremely carefully by a native speaker. We have checked again that the manuscript’s intended meaning has been retained throughout.

Reviewer 2 Report

The paper presents a design for a FlexRay-Ethernet gateway and for a security mechanism. An implementation using a FPGA is presented and tested. While the paper is interesting in principle, its scientific soundness is reduced because its structure is not appropriate from a scientific point of view. The paper contains no section dedicated to Related Work in the field, and cite a low number of references in a field which contains a lot of literature, including about integrating Ethernet for in-vehicle communication. Although the authors discuss security, the authors do not discuss elements of reliability, which is unavoidable when talking about in-vehicle communication. The authors only present results obtained in perfect working conditions, however experiments should be performed in case in which messages are not delivered reliable, and in which messages collide on the network. In a more concrete case, the diagram in Figure 6 specifies that if some all conditions are not met, the process should be restarted. On the other hand, in Table 2 the authors suggest that a high security level could be used for powertrain and x-by-wire. The authors do not discuss if in a critical situation the security model is actually usable for critical systems such as x-by-wire The authors do not approach the practical element of production costs of integrating Ethernet in vehicles, as opposed to FlexRay, as well as the cost of integrating the mentioned encryption in vehicular systems. Comparison is indeed performed against some related work, however the comparison is incorrect. Both the cited works (Kim, 2015 and Shreejith, 2017) use multiple nodes for experiments and higher network loads. This paper appears to measure only the time taken for one message to travel from one node to another. The setup is much too simplified to be relevant and to prove the reliability and performance of the system. In conclusion, although the intention of the paper is interesting, in order for it to be indeed useful to readers it needs to position itself more correctly with respect to related work, as well as to position itself more correctly with respect to the practical realities of vehicular systems. Some smaller comments: It is not clear why Figure 5 is useful. Figures 3 and 4 contain elements from previous work, which is not cited ("An introduction to FlexRay as an industrial network"). There are several English errors. The word "Costing" in the title of Section 3.3 should be replaced by "Cost".

Author Response

Responses to Reviewer’s (#2) comments:

Thank you for your time and suggestions. We have modified the manuscript based on your comments as follows.

According to your comment, we added Section 2 for Related Works, and cited more references about gateway and integrating Ethernet for in-vehicle communication. We added the description about security and reliability in-vehicle communication in Section 1 and in Section 2.2, respectively. We added the description about non-perfect working conditions, and revised the Algorithm 1 and Algorithm 2. We modified Figure 7, and added the description about that the receiving process will stop and restarted by gateway host in a more concrete case. We added the description of the security model maybe suit for critical situations in Section 4.2. We added the description of automotive Ethernet must be modified for in-vehicle networks in Section 1 and in Section 2.2. We deleted the incorrect comparison in Table 8, and deleted the incorrect sentences, and have rewritten the manuscript. We added the description of the proposed centralized gateway architecture, and the scalability of the proposed gateway structure in Section 2.1. Thank you for your time and suggestions. We revised the manuscript and added the scalability and routing mechanism of the proposed method to consider more cases. We modified Figure 5, to show what is implemented in PS and what in PL. We added the cited from related work. We have rewritten the manuscript. The content has been checked by a native speaker to correct in language usage and presentation style (English Editor: Donald Leffers). We modified the word “Costing” to be replaced by “Cost”.

Round 2

Reviewer 1 Report

I thank to authors for acceptance of most of my comments from the 1st review. Thanks to the improved quality of the paper I have understood many details and thus I have identified further minor issues and have further comments:

The main advantage of the presented GW solution is increased security. In chapter 2 some attack types are mentioned. Authors should explain in detail how their solution can avoid these attacks. In Security module description the IP, UDP and/or TCP header checks (e.g. CRC and checksums) is mentioned - it seems they are implemented outside of lwIP stack. I believe these checks are done (by default) by lwIP stack, which is a part of the GW structure. If these checks fail in lwIP, the packets are dropped. Authors should explain clearly which component of the GW does these checks. Another CRC checks are mentioned at the data-link layer (Ethernet and FlexRay frames) - here the CRC checks are always implemented in communication controllers. And again, when the check fails, the controller typically drops an incoming frame and no further processing is required. In Algorithm1 description the condition "ID is found in the routing table and security verification checks pass" is mentioned at line 6. What security checks apply here for FlexRay to Ethernet direction? In Algorithm2 description the received Ethernet frame is not decrypted (in Algorithm1 the encryption is mentioned) - why? In Figure 5 there is no CPU shown, although it is an important part of PS - why? It seems the Ethernet controller directly controls DDR memory, which is probably not true. Or the ARM cores in the Xilinx chip are not used at all? - what CPU is used to run lwIP, if this is true? Or the TCP/IP stack is implemented in PL part of a chip? I do not understand the column "Reduced" in Table 8, especially the values 93% and 89.9% in the first row. What these values are related to? The measurement of data latency should be done using higher number of nodes on both gateway sides or at least with higher traffic. Statistical results of latencies should be presented, or, in case the latency is constant under any conditions, declare this fact explicitly.

Finally I ask authors to check and improve the English language. The second paper version is much better from this point of view, but it is not good enough for publication. It is a problem for reader to understand many sentences, there are mistypes …

Author Response

The authors would like to thank the all Reviewers for their valuable comments and help. We have modified the manuscript based on the reviewers’ comments. We look forward to seeing your decision. Thank you in advance.

-----------------------------------------------------------------------------------------

Reviewer #1 comment:

I thank to authors for acceptance of most of my comments from the 1st review. Thanks to the improved quality of the paper I have understood many details and thus I have identified further minor issues and have further comments:

The main advantage of the presented GW solution is increased security. In chapter 2 some attack types are mentioned. Authors should explain in detail how their solution can avoid these attacks. In Security module description the IP, UDP and/or TCP header checks (e.g. CRC and checksums) is mentioned - it seems they are implemented outside of lwIP stack. I believe these checks are done (by default) by lwIP stack, which is a part of the GW structure. If these checks fail in lwIP, the packets are dropped. Authors should explain clearly which component of the GW does these checks. Another CRC checks are mentioned at the data-link layer (Ethernet and FlexRay frames) - here the CRC checks are always implemented in communication controllers. And again, when the check fails, the controller typically drops an incoming frame and no further processing is required. In Algorithm1 description the condition "ID is found in the routing table and security verification checks pass" is mentioned at line 6. What security checks apply here for FlexRay to Ethernet direction? In Algorithm2 description the received Ethernet frame is not decrypted (in Algorithm1 the encryption is mentioned) - why? In Figure 5

(6.1) there is no CPU shown, although it is an important part of PS - why?

(6.2) It seems the Ethernet controller directly controls DDR memory, which is probably not true.

(6.3) Or the ARM cores in the Xilinx chip are not used at all? - what CPU is used to run lwIP, if this is true?

(6.4) Or the TCP/IP stack is implemented in PL part of a chip?

  I do not understand the column "Reduced" in Table 8, especially the values 93% and 89.9% in the first row. What these values are related to? The measurement of data latency should be done using higher number of nodes on both gateway sides or at least with higher traffic. Statistical results of latencies should be presented, or, in case the latency is constant under any conditions, declare this fact explicitly. Finally I ask authors to check and improve the English language. The second paper version is much better from this point of view, but it is not good enough for publication. It is a problem for reader to understand many sentences, there are mistypes …

-----------------------------------------------------------------------------------------

Responses to Reviewer’s (#1) comments:

Thank you for your time and suggestions. We have rewritten the manuscript. The content has been checked and corrected by a native speaker.

We have added the description of the authentication scheme in Section 3.4. We have added the description of implementation of security verification in Section 3.2. and Section 3.4. We have revised the description about the processing of check fails in Section 3.2, Section 3.4, Algorithm1, Algorithm2 and Figure 7. We have added the description about the security verification of Algorithm1 in Section 3.2. We have added the description of the processing of decrypted message is in Ethernet node in Section 3.2. We have modified the Figure 5 and Figure 6, and revised the description about what is implemented in PS and what in PL in Section 3.3.

(6.1)      We have added the description about the ARM cores in PS in Section 3.3.

(6.2)      We have revised the APU and the Central interconnect access DDR memory by memory interface in Figure 5.

(6.3)      We have added the description of the ARM core (PS) used in the system, and rewritten the sentences about the implementation of Ethernet node in Section 3.3 and Figure 6.

(6.4)      We have modified the Figure 6, to clarify the TCP/IP stack what is implemented in PS and what in PL.

We have deleted the description of “Reduced” in Table 8, and added the explanation of results of latencies is constant. We have rewritten the manuscript. The content has been checked again by a native English-speaking to correct in language usage, word choice, errors in subject-verb agreement and presentation style (English Editor: James).

Reviewer 2 Report

The quality of the paper has been somewhat improved, however the previous comments on the experimental results still hold. Moreover, the results, as compared to the work of Shreejith, seem to be significantly worse, even if security has been included.

The paper should be again revised because there are English errors in the added text.

Author Response

The authors would like to thank the all Reviewers for their valuable comments and help. We have modified the manuscript based on the reviewers’ comments. We look forward to seeing your decision. Thank you in advance.

-----------------------------------------------------------------------------------------

Reviewer’s #2 comments:

The quality of the paper has been somewhat improved, however the previous comments on the experimental results still hold. Moreover, the results, as compared to the work of Shreejith, seem to be significantly worse, even if security has been included. The paper should be again revised because there are English errors in the added text.

-----------------------------------------------------------------------------------------

Responses to Reviewer’s #2 comments:

Thank you for your time and suggestions. We have rewritten the manuscript. The content has been checked and corrected by a native English-speaking.

Thank you to the reviewer. We have deleted the incorrect comparison of “Reduced” in Table 8. We have added the explanation of the results of latencies are constant under any conditions in Section 4.4. We have rewritten the manuscript of paper again, and we have had it proof-read by a native speaker to correct the grammatical errors, word choice, and pluralized for in-sentence symmetry.